# Tend and Befriend in Horses: Partner Preferences, Lateralization, and Contextualization of Allogrooming in Two Socially Stable Herds of Quarter Horse Mares

**DOI:** 10.3390/ani13020225

**Published:** 2023-01-07

**Authors:** Emily Kieson, Amira A. Goma, Medhat Radi

**Affiliations:** 1Department of Research, Equine International, Boston, MA 02120, USA; 2Department of Animal Husbandry and Animal Wealth Development, Faculty of Veterinary Medicine, Alexandria University, Alexandria 21944, Egypt; 3Department of Pest Physiology Research, Plant Protection Research Institute, Agricultural Research Center, Giza 12311, Egypt

**Keywords:** equine, welfare, affiliative behavior, laterality, friendship, stress

## Abstract

**Simple Summary:**

Recent research shows that horses create social bonds through proximity and time with limited understanding of how these friendships affect how horses respond to stress. Studies show that horses with close social connections engage in mutual grooming (allogrooming) and that this occurs more frequently during and after stress. The purpose of the study was to determine if horses interacted with the same preferred social partners during stressful events as they do during non-stressful times and if they exhibit higher frequencies in social interactions during these stressful events. The study looked at videos of 200 domestic mares in two large socially stable groups under pasture (low stress) and confined (higher stress) conditions. The videos were coded for the frequency and duration of allogrooming sessions, partner preference, and lateralization (an indicator of social and emotional processing). The results showed that there was a higher frequency of allogrooming during stressful conditions and that horses showed specific preferences for partners with whom they would allogroom. These results indicate that horses will choose very specific grooming partners during times of stress, which suggests that horses adhere to the same “tend and befriend” social stress copying strategy as other social mammals. The findings support the idea that better understanding of equine social environments and needs can more accurately assess welfare and create better environments to support equine social needs.

**Abstract:**

Studies show that horses express favoritism through shared proximity and time and demonstrate unique affiliative behaviors such as allogrooming (mutual scratching) with favorite conspecifics. Allogrooming also occurs more frequently during stress and has been observed to occur more frequently in domestic herds than feral. The role of partner preference, lateralization, and duration of allogrooming as measures of social bonding has remained unclear. The present study looked at two socially stable herds of mares (*n* = 85, *n* = 115) to determine the frequency, duration, visual field of view and partner preference during allogrooming in both pasture settings (low stress) and confined settings (higher stress). One hundred and fifty-three videos for both herds were coded for allogrooming behaviors with 6.86 h recorded in confined conditions and 31.9 h in pasture settings. Six allogrooming sessions were observed in the pasture setting with an average duration of 163.11 s. In confined settings, a total of 118 allogrooming sessions were observed with an average duration of 40.98 s. Significant (*p* < 0.01) differences were found between settings for duration (s), number of allogrooming pairs, and frequency of allogrooming (per min) for each herd. All observed allogrooming sessions involved pairs of favored conspecifics (one partner per horse). The current study suggests that horses may have friendships that can be observed through the demonstration of specific affiliative behaviors during times of stress with more frequent, but shorter affiliative interactions with preferred partners during times of stress. This context suggests that horses adhere to the “tend and befriend” principles of friendship in animals.

## 1. Introduction

Previous studies in ethology have recognized more universal applications of the term “friendship” [1,2,3] by looking at individual pairings, social bonds, and prosocial behaviors which can better inform the public about assessment protocols for positive animal welfare. Prosocial behaviors have been used to understand social behaviors in animals and are defined as engagement of an individual in actions to benefit others, specifically those that are “expected to produce or maintain the physical and psychological well-being and integrity of others” [4]. To differentiate non-human friendships from other prosocial behaviors like parental care, affiliation, sharing, social teaching, cooperation and other caring and helping behaviors [5], researchers have studied the variation in voluntary behaviors not trained responses to understand friendships between animals, including social networks and intraspecies signals [6] and under what conditions they occur.

Determining social structure, social networks, and pair bonding in horses is often based on proximity and space-sharing and the duration of time spent in the proximity of favored conspecifics or “friends” [7,8,9,10,11,12]. Researchers recognize bonded pairs based on choice and maintenance of close proximity between individuals [12,13,14,15,16]. Agonistic behaviors also occur in close proximity but result in increasing distance rather than maintaining close proximity [13].

Existing research in pair bonding in horses has focused almost exclusively on mutual grooming (allogrooming, see Figure 1) and chosen proximity. Observational studies of domestic and feral horses have shown that allogrooming is always a mutual activity between both horses [17,18,19] (where both horses actively engage in grooming) and occurs exclusively between preferred conspecifics or “friends” [7,14]. Further studies showed that allogrooming occurs more frequently during and after stressful experiences in groups of horses [10,17,18,20,21,22]. In this case, the frequency of allogrooming has been found to be greater in domestic herds compared to feral horses [13,18]. These increases in the frequency of allogrooming in a herd may suggest that allogrooming is a social coping strategy [18]. Therefore, allogrooming has been used as a means by which preferred conspecifics and equine affiliative partners can be determined [15,16,18,23] as well as a possible method of assessing both positive and negative welfare [23].

Lateralization preferences also play a role in social interactions and should be considered when observing social behaviors in nonhuman animals, especially herbivores. Stimulus emotional valence (positive or negative) and salience can determine how the animals process this information in the brain. Research has found that most species prefer to use their left side sensory organs (i.e., right hemisphere) to respond to negative valent stimuli [24]. The key area of these contralateral control of emotions is the corpus callosum, which can either execute an inhibitory influence (enhancing laterality) or an excitatory influence (diminishing laterality) on the contralateral hemisphere [25,26]. This potentially impacts the relationship between stress and lateralization [27].

Lateralization plays an important role in horse interactions as well. One study showed that horses process all their social interactions, including stressful or agonistic responses, on the left side (right hemisphere) [28] suggesting that the right hemisphere was of primary importance in processing social interactions and emotions, both positive and negative. However, in a study done by Crosby [29], it was deduced that only two horses out of nine showed a preference for left side allogrooming, while the other seven horses showed equal side preference (no side preference was reported at the population level for the behaviors studied). Laterality preference has also been found to rise with the increase in the level of concentration and task complexity [30]. Rogers [31] stated that reactivity to stress varies with laterality. These are indicated by the increased cognitive capacity gained through strong lateralized brain responses that are associated with heightened stress responses. This high degree of laterality is evidence in behavior response to tasks. Therefore, stress can alter interhemispheric integration [32], which alters the strength of lateralization and cognitive capacity. For example, horses show high escape responses when they receive negative stimuli with the left eye [33]. This means that attentional state, emotional reactivity, and competitive interactions are high when perceived from the left side [34]. This left side reaction to negatively valent stimuli is applied to management and husbandry contexts. Therefore, lateralization in affiliative interactions could be an indicator of the degree of horse attention and emotional experience, not the nature of attention or emotion. A better understanding of affiliative behaviors and lateralization between pairs and the contexts in which they occur could help to support improvement in management practices and welfare.

The objective of this study was to study the allogrooming behaviors of two different socially stable herds of mares under different conditions to determine if partner preference, frequency, lateralization, and duration changed between conditions. This would then provide information to the specific indicators of changes in affiliative responses. This objective was fulfilled through determining the frequency, duration, field of vision (lateralization) and partner preference during allogrooming under low-stress conditions (pasture) versus elevated stress conditions (group confinement). The hypothesis of this study was that stress of confinement can lead to an increase in the frequency and duration of allogrooming with a specific partner since allogrooming occurs between bonded or paired partners. Therefore, these affiliative allogrooming behaviors can be used as a means for improving indicators of positive psychological welfare in horses.

## 2. Materials and Methods

The videos involved in this study were collected with the approval of the Oklahoma State University IACUC no. AS-17-5.

### 2.1. Subjects and Management

The observations were done at the Noble Equine Center, a private quarter horse veterinary and breeding facility in Purcell, Oklahoma, USA that houses approximately 500 horses. The owners and managers of the facility consented to allow the researchers to observe and record the behaviors of horses during normal daily operations. Of these horses, two large herds of Quarter horse mares (*n* = 200) were utilized. One herd consisted of 85 mares while the other consisted of 115 mares. The videos were recorded three days a week (weather permitting) during the period from March to May in 2019. Videos were collected from 09:00 until 15:00 each day during this period. The property encompasses approximately 160 acres (65 hectares) split into multiple pastures enclosing around 20 to 40 acres each. The space allowance in pasture was 703.65 to 1407.30 m^2^ per horse. The age of the mares ranged from 6 to 20 years, and all had been at the facility for at least 2 years living in stable social groups. All horses live permanently in large outdoor pastures (herds are rotated through a series of fenced pastures throughout the year but herds remain intact) with free access to hay/grass (bermuda grass and native prairie grass) and water. The larger areas allow horses to exhibit spatial preferences within the herd and express choice of proximity with other individual members of their herd.

As part of the normal daily routine of the facility, each herd was brought into two small pens of approximately 0.81 hectares (space allowance per horse was 70.43 m^2^) each, closer to the large palpation/insemination barn at 10:00 to 11:00 am every day (the herds were kept separate) to separate the mares that required palpation to determine ovulation. Mares that needed palpation were separated from the herd while the remaining horses were let back into the pastures. After palpation, individual mares were reunited with their respective herd. The duration of their stay in the confined areas ranged from 20 min to one hour. The total duration of individual handling by humans was approximately ten minutes every second or third day.

In the pasture settings, horses had free access to grazing or multiple hay stations as well as water and they were also able to spread out and freely choose proximity to conspecifics and find shelter as needed. Due to large areas and open access to plentiful resources, stressful conditions were minimized. The confined settings (the smaller pens) had limited space, as mentioned above, no hay and restricted water access (one water trough per pen and up against a fence), which was conditional based on where the horse was and the number of horses in the pen. Conditions of the smaller pens, therefore, represented the restricted conditions. Observers for horses under these conditions witnessed additional stress behaviors that included increased movement, muscle tension, agonistic behaviors associated with desire for increased spatial needs and vigilant behaviors indicative of higher stress responses [35,36], further indicating that the smaller pens created more stress responses in horses.

### 2.2. Videos Collection

Videos were recorded using 14 GoPro cameras (Hero model number CHDHA-301, GoPro Inc., San Mateo, CA, USA, purchased used on Ebay). Eleven cameras were fixed on fences of the pastures containing horses, while the three other cameras were positioned around the smaller pens that are closer to the large palpation/insemination barn. Cameras were positioned to record all areas of the pastures and pens and all horses within each area at time of recording.

A total of 185 videos for both herds were recorded. Of these videos, 153 videos contained footage of horses of which 33 videos were of horses in confined (smaller pen) settings and 120 videos in pasture settings. The total useable video footage was 4782.11 min (79.70 h). For the confined settings, the videos length was 411.64 min (6.86 h) with a mean of 12.47 min/video that were observed and coded. For pasture settings, the length of videos was 1915.08 min (31.9 h) with a mean of 15.96 min/video that were also coded. For Herd 1, both settings were 94 videos of 1553.54 min (25.89 h) in length with a mean of 14.66 min/video. For Herd 2, both settings were 59 videos of 901.85 min (15.03 h) with a mean length of 15.29 min/video (Table 1).

Three researchers coded all the videos that were collected by focal sampling observation independently. The allogrooming behavior was defined according to Crosby [25]. An allogrooming session was recorded when one horse positioned itself to the side of the other and started grooming a particular area and the other horse reciprocated. If one horse started grooming the other horse, but the action was not reciprocated, the activity was not recorded as an allogrooming session. Once grooming stopped, then a separate occurrence was recorded. Each observer counted the total number of allogrooming sessions that occurred during each video, the duration of each allogrooming session, the number of pairs that engaged in allogrooming (as some pairs engaged in allogrooming more than once in the same video), the number of partners with whom each horse engaged in allogrooming, as each pair performing allogrooming was tracked by subsequent footages from video, and the visual field of view with which the horses initially engaged in allogrooming.

### 2.3. Statistical Analysis

Once all videos were coded, total counts for the number of allogrooming sessions for each video were combined and averaged across three observers to come up with a final count. In the case of duration, numbers were averaged within each observation and then an average of averages was determined to come up with the final duration per video. A consensus was reached for each video for data concerning partner preference and lateralization. Each variable was coded and analyzed separately and averages and consensus for each variable were determined independently (therefore, total number of lateralization counts differed from total counts). Independent t-tests and chi-square tests were applied for settings where variables were available. In addition, a correlation matrix was conducted to determine the relation between parameters measured and their dependency on each other. Statistical analysis was done through the use of MS Excel (2022).

## 3. Results

Overall, allogrooming (Figure 1) appeared in 33 of these videos with 462.13 min length (7.70 h): 29 videos of confined settings with a length of 388.55 min (6.46 h) and the other 4 videos in pasture settings of 73.58 min (1.23 h); see Table 1 for more details.

### 3.1. Allogrooming Measurements in Confined Settings

The descriptive data for the footage of horses in confined settings, in which allogrooming appeared, are shown in Table 2. The total number of allogrooming sessions observed in confined settings was 118. The average duration of each allogrooming session in confined settings was 40.98 s. The total number of left allogrooming pairs was 66 and for the right was 71. All but one observed allogrooming interactions involved horses with a single partner (horses did not switch partners except one horse in one video).

### 3.2. Allogrooming Measurements in the Pasture Settings

Only Herd 1 had videos in pastures setting in which allogrooming occurred. There were no videos in the pasture for Herd 2 in which allogrooming occurred. The total number of allogrooming sessions observed was 6. The average duration of each allogrooming session was 163.11 s. The total number of left allogrooming pairs was 3 and the right side allogrooming was similar in number. All allogrooming interactions involved a single partner (Table 2).

### 3.3. Comparison between the Two Settings in Allogrooming Measurements

Independent *t*-tests were done to compare the two settings for the different allogrooming measurements (Table 3). There was a clear significant difference for both herds between the two settings with regards to the total number of single partner affiliative pairs for each setting (*p* < 0.001) and for average duration of allogrooming sessions (*p* < 0.001). There was also a significant difference in duration of allogrooming sessions between settings for Herd 1 (*p* < 0.01). There was no allogrooming appearance in pasture for herd 2, so duration data could not be compared to the confined setting.

Independent *t*-tests were also done to compare allogrooming frequencies (number of allogrooming sessions per minute) between confined and pasture settings for all horses and for Herd 1 and Herd 2 separately (Table 4). There was a clear significant difference for both Herd 1 (*p* < 0.01) and Herd 2 (*p* < 0.001) between the two settings with regards to the frequencies of allogrooming. When herd data were combined for each setting, there was a significant difference between the frequency of allogrooming in confined and pasture settings (*p* < 0.01).

Frequencies of allogrooming were also compared between Herd 1 and Herd 2 for each setting to determine any significant differences (Table 5). No significant differences were found between the herds for either confined or pasture settings.

Independent *t*-tests were also done to compare allogrooming lateralization (right or left side preference for allogrooming) between confined and pasture settings for all horses and for Herd 1 and Herd 2 separately (Table 6). There was no significant difference between the two settings with regards to the side preference of allogrooming in either setting for either Herd 1 or Herd 2 (no allogrooming data is available for Herd 2 in pasture settings). When herd data were combined for each side preference (left or right), there was no significant difference between confined and pasture settings for left (*p* = 0.092) or right (*p* = 0.261) side preference in allogrooming.

Left and Right lateralization (side preference) of allogrooming were also compared for each setting for Herd 1 and Herd 2 to determine any significant differences (Table 7). No significant differences were found between side preferences for the herds for either confined or pasture settings.

### 3.4. Correlation between the Allogrooming Measurements

When studying the correlation between the allogrooming measurements of the frequency (allogrooming sessions per min) to duration (s) and lateralization preference in each video (Figure 2, Figure 3 and Figure 4), a positive correlation was found between the frequency (M = 0.346, SD = 0.321) and the duration (M = 41.587, SD = 27.049) of allogrooming in confined settings (Figure 2) with r^2^ = 0.460 (*p* < 0.001) (one outlier was removed). There were only four videos in the pasture setting in which allogrooming occurred, so no correlation coefficient was calculated for the pastures setting separately. Correlations between frequency (M = 0.329, SD = 0.317) and duration (M = 42.195, SD = 26.582) for all horses in all settings was also observed (Figure 3) with r^2^ = 0.494 (*p* < 0.001). This means the duration increased with the increase in the frequency of allogrooming in confined settings and all settings. There was a positive correlation also between left side allogrooming and duration with r^2^ = 0.596 (*p* < 0.001). Additionally, the number of pairs was correlated positively with left side allogrooming with r^2^ = 0.880. (*p* < 0.001). Similarly, the same correlations were observed with the right side allogrooming with r^2^ = 0.534 (*p* < 0.001) and 0.909 (*p* < 0.001), respectively. This means no side preference correlated with the number of pairs or duration. While there was no significant difference between preferred lateralization of initiated allogrooming, there were instances where horses changed sides during allogrooming sessions. There was a positive correlation between the frequency of allogrooming and the number of allogrooming sessions in which visual field of view changed in confined settings with r^2^ = 0.277 (*p* < 0.01) (Figure 4). There was not enough data to determine the correlation between measures in the pasture setting. In addition, a chi-square test of independence was performed to look at the relationship between lateralization and setting. The relationship between these variables was not significant, *X*^2^ (1, *n* = 143) = 0.0077, *p* = 0.93023.

## 4. Discussion

Overall, the findings of this study indicate that horses deliberately engaged in mutual grooming (allogrooming) interactions with increased frequency with specific preferred partners during stressful conditions, suggesting that this is not only a stress response but also a specific targeted social behavior between bonded partners. Allogrooming pairs also displayed significantly longer duration of allogrooming sessions in pastures settings versus confined settings despite having significantly less frequent displays of allogrooming behavior.

### 4.1. Partner Preferences

Out of the 124 observed allogrooming sessions, only one horse was observed partaking in allogrooming with more than one partner, suggesting that these horses had preferred conspecifics with whom they would engage in this specific affiliative behavior. This echoes what has been seen in other studies regarding partner preferences of horses and willingness to engage in affiliative behaviors with select conspecifics [7,8,9,10,11,16]. The significant increase in frequency with preferred partners (and no changes in partners) under confined conditions in which stress behavior was observed suggests that these confined conditions impacted the frequency in which these horses engaged in partner-specific affiliative allogrooming.

The increased frequency of allogrooming in confined settings aligns with previous research in horses that suggests that allogrooming is used as a coping strategy for horses in domestic settings [18]. In behavioral psychology, the deliberate initiation of more frequent and more intense social affiliative interactions with preferred partners during stressful conditions has been witnessed in humans as well as other animals [37,38,39,40,41]. This response, known as the “tend and befriend” response, has been recognized as a social response to stress in which individuals seek affiliative interactions with preferred partners more frequently or intensely during and after stressful events [37,38].

In the “tend and befriend” response, one or more individuals respond to stress by seeking out familiar friends and engaging in highly affiliative pro-social strategies at a higher intensity and/or frequency for a short time [37,38,39,42,43]. Such an increase in pro-social behaviors after the stress has also been seen in laboratory experiments with rats [39,40,41]. The findings of this study indicate that horses also engage in the “tend and befriend” response when under stressful conditions.

### 4.2. Number of Pairs, Frequency, and Duration

There were significant differences in number of pairs, frequency and duration of allogrooming sessions between the confined and pasture settings. The number of grooming pairs and the frequency of allogrooming sessions were significantly higher in the confined settings versus pasture settings whereas the duration of each allogrooming session was significantly higher in pasture settings versus confined settings.

Research shows that psychological welfare could be improved when horses are housed in group environments [44,45,46] and that the frequency of social behaviors decreases with lower stocking density [47]. This could be linked to more time and space to engage in grazing, resting, and other natural behaviors if resources are restricted in confined areas. Agonistic behaviors have also been seen more frequently in groups of horses that are socially unstable [48], suggesting that higher stocking densities and smaller spaces result in higher frequency of social interactions. Furthermore, agonistic behaviors have been linked to space sharing and choices of closest conspecifics [13,16,49,50], suggesting that both agonistic and affiliative behaviors may increase in more confined areas and higher stocking density.

The findings of the correlations further support the increase in frequency and activity in confined areas and in combined data from both settings. The correlations resulted in clear trends of increase of duration of allogrooming with an increase in frequency. In confined settings where horses displayed stress behaviors, current research suggests that more allogrooming sessions would be observed. If there is no disruption to these expressions, then they could perhaps be allowed to continue in duration. It is possible that in circumstances where allogrooming occurred with more frequency, they were also ideal conditions for horses to continue with each allogrooming session, thereby creating a positive correlation between frequency and duration.

When comparing settings, lower duration of allogrooming sessions in confined environments versus pasture settings could be a result of the observed increase of movement (due to higher stocking density and higher prevalence of stress behaviors leading to more movement) which could also decrease the ability of a single pair to stay stationary in order to continue allogrooming activities. Although many of the pairs were observed ending the allogrooming session of their own volition, the factors involved in decreased duration of allogrooming in confined settings deserves further exploration.

### 4.3. Lateralization

The results further showed that there were no distinct lateralization preferences for horses during allogrooming activities. Other studies indicate that horses have a lateralization preference for processing social stimuli. Lateralized behavior demonstrated by sensory laterality is considered the clearest form for the expression of hemispheric specialization [51] with laterality strength also playing an important role, sometimes even greater than direction [52]. Rogers [31] stated that the strength of lateralized responses range from weak to strong laterality or absence. However, weak or absent lateralization does not mean less cognitive processing but means the involvement of one hemisphere alone in a particular behavior (strong laterality), or contribution of the other hemisphere to some degree. Several studies showed that each hemisphere is involved in performing certain tasks. This reflects that lateralization increases cognitive capacity. In this study, there was no significant difference between preferred lateralization of allogrooming in either herd or in either setting. The correlation of changes in lateralization with frequency resulted in a small positive correlation. Much like the correlations between frequency and duration, this could be a result of increased ability (and environmental conditions) to express desired behaviors with a preferred partner.

Although individual lateralization can increase neural efficiency and allows multitasking [53,54], it has disadvantages for individuals when the stimulus appears equally on either body side [55]; also, detecting a predator can take a longer time on the non-preferred side, as in toads [56]. These visual lateralization examples are considered disadvantageous in horses too, as their eyes are laterally positioned with a binocular vision of an angle of 60–70° in the front, a monocular vision of 205° on the sides and a blind spot behind [57].

Despite physical disadvantages, there is still clear evidence of lateralization preferences with horses when given choice of interaction with new stimuli. The act of allogrooming does not seem to fall into this category, suggesting that the act itself is more important than the process of incoming social signals [28,29]. This might then also suggest that the initiation of allogrooming behavior and the lack of lateralization preference in this study is much more focused on the initiation of the act and mutual exchange of stimuli rather than the processing of incoming social signals. Similar findings from the Crosby [29] study corroborate this suggestion. He deduced that only two horses out of nine showed a preference for left side allogrooming, while the other seven horses showed equal side preference (no side preference was reported at the population level for the behaviors studied). The findings of this study support his research.

### 4.4. Welfare

Understanding affiliative and emotional responses of horses in domestic settings has the potential to influence management styles and improve equine welfare [58,59,60]. Recognizing that horses may choose to display specific affiliative behaviors with unique partners can provide insight into psychological welfare. A better understanding of the role of affiliative behaviors in different spaces can serve to support development in management practices, especially as it pertains to individualized expressions and an improved understanding of when, how and what behaviors they choose to express and with whom they choose to show them. Recognizing when affiliative interaction occurs, the specifics of the interactions and the context of these interactions presents a means of understanding and addressing welfare concerns. Social animals need social interaction as part of their welfare needs [5,31,61,62]. Therefore, understanding the role of affiliative interactions during stress and the role of affiliative activities in positive welfare can help us better understand how to assess welfare and address needs when they are not met.

### 4.5. Limitations

The study has limitations based on the nature of the breed and social conditions under which the horses were recorded. The study location was limited to one site with only mares, so further investigation is needed at other sites involving stable social groups of mixed genders, including geldings and stallions if possible, to determine other factors that might play a role in allogrooming. Additional studies should also be done on free-living and feral herds to determine if allogrooming occurs more frequently during and after stressful events with specific partners. Additionally, as more information is gathered regarding affiliative behaviors, additional research is needed to determine if other affiliative behaviors occur more frequently as a result of stress.

## 5. Conclusions

The study found that allogrooming behaviors between socially bonded horses in socially stable herds increase in frequency and duration in confined environments where other stress behaviors are observed. There was no significant difference between right and left allogrooming (lateralization) but there was an increase in duration of allogrooming sessions in pasture (lower stocking density) environments and a positive correlation between frequency of allogrooming and duration. The findings suggest that horses deliberately engage in affiliative social grooming in higher frequency with bonded partners in environments where other stress behaviors are observed. This increase in affiliative behaviors with specific partners can help inform better management and husbandry practices with regards to positive welfare and wellbeing. Further studies need to determine if humans (or other animals) may be able to fill this need for horses who may not have horse companions with whom they choose to engage in affiliative behaviors. Additional studies need to be done to determine if additional affiliative strategies are used as part of social coping mechanisms.

## Figures and Tables

**Figure 1 animals-13-00225-f001:**
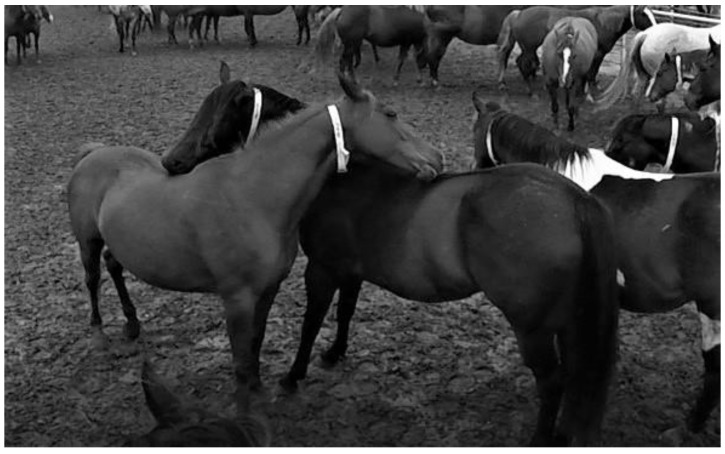
Left allogrooming in an affiliative pair.

**Figure 2 animals-13-00225-f002:**
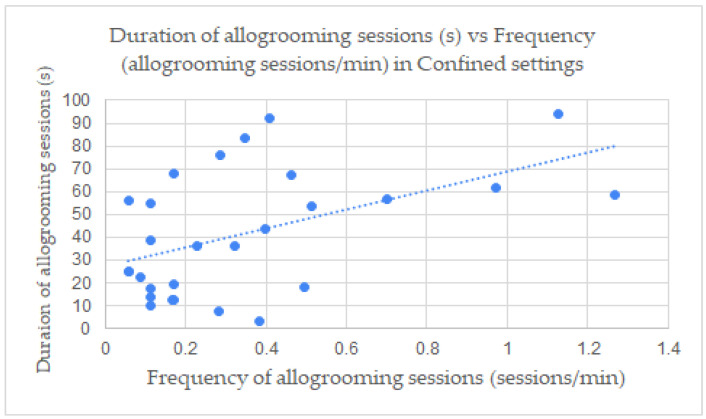
Correlation between frequency (allogrooming sessions/min) and duration (s) of allogrooming session in confined settings. r^2^ = 0.460612 (*p* < 0.001); r^2^ = 0.460612 (*p* < 0.001).

**Figure 3 animals-13-00225-f003:**
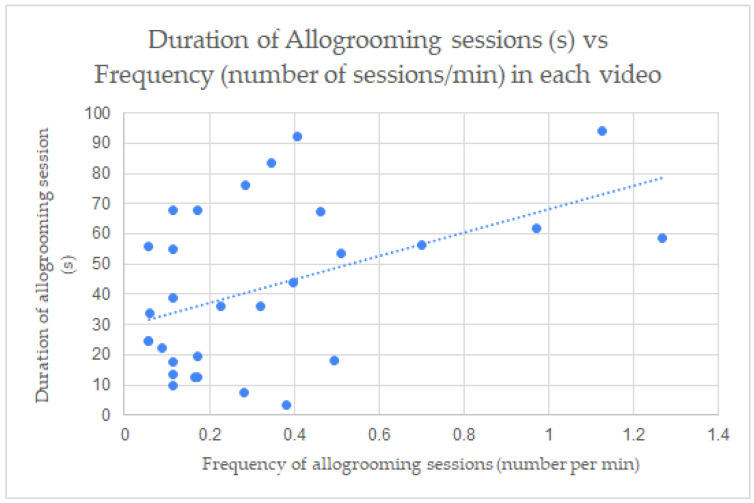
Correlation between frequency (allogrooming sessions/min) and duration (s) of allogrooming session in combined confined and pasture settings (all videos). r^2^ = 0.494 (*p* < 0.001).

**Figure 4 animals-13-00225-f004:**
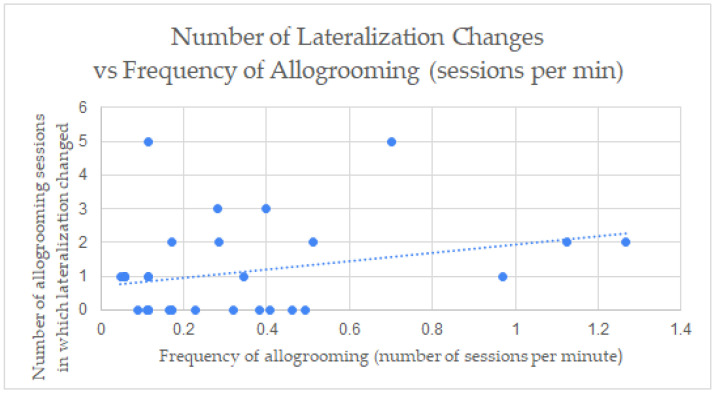
Correlation between frequency (number of allogrooming sessions/min) and number of allogrooming sessions in which visual field of view changed in pasture and confined settings. r^2^ = 0.277 (*p* < 0.01).

**Table 1 animals-13-00225-t001:** Total number of recorded videos, allogrooming videos and the length of these videos in designated herds and setting.

Videos	Number	Length
Total videos	185	33 min/video
No horse and/or undesignated herds	32	Excluded from Study
Total videos for herd 1 and 2: confined + pasture settings	153	Total: 4782.11 min
Combined videos for Herd 1 and 2: confined setting	33	Total: 411.64 min, Mean: 12.47 min
Combined videos for Herd 1 and 2: pasture setting	120	Total: 1915.08 min, Mean: 15.96 min
Total videos for Herd 1: confined and pasture settings	94	Total: 1553.54 min, Mean: 14.66 min
Total videos for Herd 2: confined and pasture settings	59	Total: 901.85 min, Mean: 15.29 min
Total videos for Herd 1: confined setting	21	Total: 282.97 min, Mean: 13.47 min
Total videos for Herd 1: pasture setting	73	Total: 1141.90 min, Mean: 15.64 min
Total videos for Herd 2: confined setting	12	Total: 128.67 min, Mean: 10.72 min
Total videos for Herd 2: pasture setting	47	Total: 773.18 min, Mean: 16.45 min
Total number of videos in which allogrooming appeared	33	Total: 462.13 min
Total number of of videos in which allogrooming appeared for Herd 1 and 2: confined setting	29	Total: 388.55 min, Mean: 13.03 min
Total number of of videos in which allogrooming appeared for Herd 1 and 2: pasture setting	4	Total: 73.58 min, Mean: 18.39 min
Total number of of videos in which allogrooming appeared for Herd 1: confined setting	18	Total: 262.29 min, Mean: 14.57 min
Total number of of videos in which allogrooming appeared for Herd 1: pasture setting	4	Total: 73.58 min, Mean 18.39 min
Total number of of videos in which allogrooming appeared for Herd 2: confined setting	11	Total: 126.30 min, Mean 11.48 min
Total number of of videos in which allogrooming appeared for Herd 2: pasture setting	0	0

**Table 2 animals-13-00225-t002:** Descriptive data for the frequency, number of affiliative pairs, average duration, number of affiliative partners in pairs and left/right visual field of view of allogrooming in different settings.

Item	Total	Confined Environment (Higher Stress)	Pasture Environment (Low Stress)
Total number of allogrooming sessions	124	118	6
Average Frequency of allogrooming sessions (per minute of video)	0.124	0.563	0.003
Number of affiliative pairs allogrooming	124	118	6
Average duration of individual allogrooming sessions	55.784	40.98s	163.11s
Number of partners in affiliative interactions	1	1	1
Number of allogrooming sessions with single partner	123	117	6
Number of horses with two or more partners	1	1 (One horse with three partners in one video)	0
Total Number of pairs engaged in left allogrooming	69 *	66	3
Total number of pairs engaged in right allogrooming	74 *	71	3

* Totals for lateralization were calculated separately and may not match total allogrooming count.

**Table 3 animals-13-00225-t003:** Independent *t*-test comparisons between confined and pasture settings in duration (s) and number of allogrooming pairs (M, SD, and df are per video observation).

		Confined			Pasture				
Item	Total	Mean (M)	SD	df	Mean (M)	SD	df	T-Value	*p*-Value
Duration (s) of allogrooming sessions	1840.86	40.980	26.762	28	163.108	150.737	3	2.744	<0.001 **
Duration (s) of allogrooming sessions in Herd 1	1436.11	43.538	24.938	17	163.108	150.737	3	2.845	0.003 *
Duration (s) of allogrooming sessions in Herd 2	404.75	36.795	30.284	11	N/A	N/A	N/A	N/A	N/A
Total number of pairs allogrooming (single partners)	124	3.576	4.486	32	0.050	0.219	119	2.609	<0.001 **
Number of pairs allogrooming in Herd 1 (single partner)	93	4.143	5.369	20	0.050	0.219	72	2.630	<0.001 **
Number of pairs allogrooming in Herd 2 (single partner)	33	2.583	2.109	12	0.000	0.000	47	2.665	<0.001 **

* *p* < 0.01, ** *p* < 0.001.

**Table 4 animals-13-00225-t004:** Independent *t*-test comparisons between confined and pasture settings in frequency (number of allogrooming sessions per minute video).

	Confined			Pasture				
Item	Mean (M)	SD	df	Mean (M)	SD	df	T-Value	*p*-Value
Herd 1	0.716	1.905	21	0.005	0.020	72	2.630	0.002 *
Herd 2	0.295	0.255	12	0.000	0.000	46	2.665	<0.0001 **
Combined (Herd 1 and Herd 2)	0.563	1.528	32	0.003	0.016	119	2.609	<0.0001 **

* *p* < 0.01, ** *p* < 0.001.

**Table 5 animals-13-00225-t005:** Independent *t*-test comparisons between Herd 1 and Herd 2 in frequency (number of allogrooming sessions per minute video).

	Herd 1			Herd 2				
Item	Mean (M)	SD	df	Mean (M)	SD	df	T-Value	*p*-Value
Confined	0.716	1.905	21	0.295	0.255	72	2.733	0.454
Pasture	0.005	0.020	12	0.000	0.000	46	2.618	0.129

**Table 6 animals-13-00225-t006:** Independent *t*-test comparisons between confined and pasture settings in lateralization (number of left or right side preference for allogrooming in each video).

Item	Confined			Pasture				
	Mean (M)	SD	df	Mean (M)	SD	df	T-Value	*p*-Value
Left allogrooming total	2.2	2.024	29	0.600	0.548	4	2.733	0.092
Right allogrooming total	2.367	3.409	29	0.600	0.548	4	2.733	0.261
Herd 1 Left allogrooming	2.526	2.270	18	0.600	0.548	4	2.819	0.077
Herd 1 Right allogrooming	2.789	4.158	18	0.600	0.548	4	2.819	0.260
Herd 2 Left allogrooming	1.636	1.433	10	N/A	N/A	N/A	N/A	N/A
Herd 2 Right allogrooming	1.636	1.286	10	N/A	N/A	N/A	N/A	N/A

**Table 7 animals-13-00225-t007:** Independent *t*-test comparisons between Right and Left lateralization for Herd 1 and Herd 2 in each setting (number of left or right side preference for allogrooming in each video).

Item	Left Allogrooming			Right Allogrooming				
	Mean (M)	SD	df	Mean (M)	SD	df	T-Value	*p*-Value
Herd 1 Confined	2.526	2.270	18	2.789	4.158	18	2.719	0.810
Herd 1 Pasture	0.600	0.548	4	0.600	0.548	4	3.355	1.000
Herd 2 Confined	1.636	1.433	10	1.636	1.286	10	2.845	1.000
Herd 2 Pasture	N/A	N/A	N/A	N/A	N/A	N/A	N/A	N/A
Both Herds Confined	2.200	2.024	29	2.367	3.409	29	2.663	0.819
Both Herds Pasture	0.600	0.548	4	0.600	0.548	4	3.355	1.000

## Data Availability

Please contact the primary author regarding data involved in this study.

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
