# Peer review of "Tend and Befriend in Horses: Partner Preferences, Lateralization, and Contextualization of Allogrooming in Two Socially Stable Herds of Quarter Horse Mares"

_animals, 2023, doi:10.3390/ani13020225_

Round 1

Reviewer 1 Report

Overall the article provides some insight into very specific behaviors observed in domestic mares. However the study has limitations, which the authors addressed in the discussion. The introduction is well written and easy to understand, with appropriate citations. More details are needed in the materials / methods section. The presentation of the results could be written with more clarity and the authors need to avoid redundancy. 

Author Response

Thank you for your review.  Please see the attachment for responses.

Reviewer 2 Report

This is an interesting study that will contribute to our understanding of how domestic horses may engage in prosocial behaviours during periods of behaviour restriction and interestingly that increases in allogrooming during confinement may be indicative of a behavioural stress response.  The study does have a number omissions in regards to the description of the method, data analysis and the discussion that require revision before the study would be suitable for publication.

General comments:

The introduction is well written and provides a suitable background to the study.

The methods, results and discussion sections require revision.

What constitutes an allogrooming bout is not defined. On what basis was a bout considered to have commence-concluded?  Horses engaged in allogrooming may groom for a period, then cease briefly and then recommence grooming, sometimes in a different location (e.g. switch from wither to rump area).  An ethogram of what constitutes allogrooming and how a bout was defined/counted citing the relevant literature needs to be provided.

Not enough information is provided regarding the videos.  For example, on how many days were videos recorded?  Is the duration of recording equal across days?  At what time were the videos of the pasture setting taken?  Did the videos of the pasture setting capture all horses in the herd?  Where these data captured at the same time as the confined setting?  Horse behaviour follows a diurnal rhythm with periods of resting, grazing, affiliative interactions occurring throughout the day.  One possible reason for the lack of allgrooming observed in the pasture setting was that horses were engaged in other activities such as grazing.  The authors have not cited references on rates of allogrooming in pastured horses generally which would provide greater context to whether the lack of allogrooming in the pastured horses represents normal equine behaviour. The time period in which the videos were captured for all herds/treatments needs to be stated. 

The presentation of the results as “per video” is not robust as the videos were of different lengths and it appears, taken over several recording sessions per treatment group and herd.  A more robust way to present the results of frequencies and durations of allogrooming would be to report them as per minute or per hour of footage.  The metric per video is not relevant unless the videos were of identical duration which they were not.

More information needs to be provided in regards to the stress state of the horses.  It is simply assumed that the horses experienced the confined setting as a stressor, but further evidence of this needs to be provided or the relevant literature cited to substantiate this.  It is stated that the horses undergo handling in the confined setting on a routine basis it would be expected that there is a degree of habituation to the setting and consequently, the horses may not experience the setting as a stressor.  Consequently, you need to provide more evidence that the horses experienced this setting as a stressor given no physiological data of stress system activation is provided.  .  As video data of behaviour have been captured, behavioural evidence of stress using validated indicators should have been provided as well as behavioural indicators of a lack of stress in the pastured horses.   

The discussion is incomplete.  There is no discussion of the lack of allogrooming observed in the pasture settings- the possible reasons for this lack needs to be provided as these horses served as a control.  Alternatives to the tend-befriend hypothesis should have been considered.  The discussion requires a more detailed examination of the results and alternative hypotheses and a more detailed examination of relevant literature in relation to the findings presented here, including alternative explanations for the results- e.g. differences between the two groups arising from differences in opportunities to perform the full suite of behaviours in the confined setting, and the close proximity of conspecifics compared to the pasture (representing an opportunity to engage in the behaviour in the absence of alternative behaviours such as grazing) rather than only being a stress coping response.

Specific comments on the text

Numbers refer to line number in text

26-42 Include means or ranges for durations of recordings/allogrooming

84.  Please provide a reference for this statement re emotional processes in the brain as well as more detail in regards to which regions you are referring to as emotional processing in the brain occurs across a range of regions via very complex processes in areas such as the amygdala, basal ganglia, hippocampus and frontal cortices as well as other regions.  

90.  It is not clear what “laterality strength” means in this sentence-please explicate further.

99-100. Explain further here what is meant by “task complexity” -what kind of tasks and what is the relevance of this to the study design?

112-118.  Long sentence that is hard to follow.  Suggest rewording into several smaller sentences to improve the clarity.

118.  “Need” used twice in the sentence.

165-174.  Over how many days/sessions was the video collected?  This needs to be clearly stated.  If collected over several sessions per herd this needs to be stated and statistical analysis into whether there were differences in allogrooming rates between these sessions within each treatment needs to be conduced.  If the videos were collected within a single session for each herd/treatment this needs to be stated- e.g. all videos for each herd/treatment were collected in a single session or similar.

Time of day videos were collected- this needs to be more precises than “around 10am” for the confined setting horses- e.g. between X AM and X AM for each setting-confined and pastured so it is clear that either, the same period for both groups was recorded, or different time periods were used. If different time periods, the potential contribution of this to the results (see general comment above re time diurnal rhythm of equid time budgets above).

Did the videos cover all areas of the two spaces- that is, where there areas within the spaces in which behaviour occurred that was not captured by the cameras and consequently couldn’t be analysed? Please clarify.

In addition, please describe how an allogrooming bout was defined, referring to the relevant literature for validation of definition (e.g. ethogram) and the criteria used to determine when a bout commenced and when it concluded.

173. What is “visual field of view” listed here- provide an explanation.

182. T-tests are used-was the data checked for normality which is a requirement for this test-please add.  Please add the statistical significance level (p>0.5?)

184.  Over how many days/sessions were videos taken? 

183-195.  The mean of videos is provided but it is not clear

196. Table 1.  This table, while detailed lacks critical information.  It is not clear over how many days the videos were captured.  Do these data represent a single recording session for each of the two herds? If yes, then the means of videos are appropriate.  If captured over several sessions per herd these data need to be provided and any differences between sessions should be analysed as a factor.  Based on the number of videos collected it appears that recording has occurred over multiple sessions. 

198. Remove table caption

221. Table 2

The use of averages per video is not valid as the videos are of differing lengths and multiple cameras were used to capture the data.  These data should be reported as per minute or per hour of footage to provide a more accurate representation of the frequency of the activity.

224. Always include the test, test statistic and p-value whenever referring to statistical results-in this case, refer to the table in which the statistics are quoted.

226.  Refer to table to guide readers to the location of the statistical test

236. Include p value with correlation coefficient here and elsewhere.

237. Don’t use abbreviations (e.g. cc) without defining first.

247. Include r value on graphs

256. Unless further behavioural evidence is included to validate the horses experienced the confided setting as a stressor, remove the word “stressful” here, replace with “confined” and modify along the lines of “which was likely to lead to stress or increased anxiety” or similar. 

257. You have not made the case here for why allogrooming is a coping strategy, particularly because you have not demonstrated that the confinded environment did cause stress system activation or a change in affective state (e.g. anxiety, frustration, fear) compared to the pasture setting.  This needs to be more cautiously stated or the discussion expanded with more detailed reference to the relevant literature in horses or other suitable species.  What benefits could the allogrooming actually provide?  The literature on allogrooming in horses (at least when conducted in a human-horse dyad) is that it can induce beneficial physiological effects such as reduced heart rate and reduced behavioural indicators of arousal.  This needs to be considered in the context of the “tend and befriend” hypothesis discussed here.

271. Repetitive.

281-282. Please expand this further- for example, what specifically did Crosby find that is relevant here?

No consideration of the control group- please discuss possible reasons for the lack of allogrooming in the control condition-what were the horses doing instead of allogrooming?

Please also consider allogrooming in the confined setting in the context of the beahvioural restriction (i.e, horses could not graze during the confinement), so was the increase in frequency a displacement activity in the absence of alternative options?

285. Site, not sight.

296.  As per previous comments re the lack of evidence provided that the horses experienced the confinement as a stressor.

Author Response

(The authors gave the same response as above.)

Reviewer 3 Report

Dear authors

Thank you so much for your work. I revised the manuscript entitled “Tend and Befriend in Horses: Preferences, Lateralization, and Contextualization of Allogrooming in Two Socially-Stable Herds of Quarter Horse Mares”. It is an interesting topic but it presents several flaws – English, methodology, ethical statement, presentation of results and discussion – that must be addressed before to consider the present manuscript for publication.

Generally, I suggest to revise your paper for English since some sentences are difficult to follow and can lead to misunderstandings.

Introduction

The introduction is too long and should be shortened. Moreover, there are some statements or speculations that are not supported by adequate references since refereed to other animal species. This creates lot of confusion and mistakes of interpretation of the current scientific literature. Moreover, it is not clear which is the novelty of the present study and –

accordingly – the aims are confusing.

Generally, aims of the study are not clear. The authors should be more concise. Moreover, they should add which is their hypothesis. Lines 128-130 seem to describe a sort of practical implication that is really general. I appreciate the fact that authors underline the practical implications of the study but this should be move in the discussion and conclusion and must be specific and related to the obtained results.

Methods.

Ethical statement – the approval was related to the collection of videos. Anyway, there is not mention to the fact that the smaller pens with restricted conditions were characterised by limited space, no hay and restricted access to water… this aspect must be clarified from an ethical point of view since represent a main welfare concern!

In the paragraph related to statistical analysis there is no mention on the correlation analysis described in the paragraph 3.4.

Results:

There is confusion in Table 1 probably because there is not a congruence between the definition used. I suggest to uniform the definitions according to the different type of herds. An example “horses in herd 1 and 2 both setting videos” could be “total number of videos collected from herd 1 and 2: confined setting + pasture setting”.

Discussion

The discussions are too speculative and should be fine-tuned on study results. I recommend reviewing and beginning the discussion with the answer to the aim in the introduction, immediately followed by the evidence presented in the results that corroborate it. Many results have not been properly contrasted and are speculates with generalizations that are not very conducive to obtaining valid conclusions and recommendations from the results obtained. Therefore, also the conclusions need to be revised on the basis of practical implication of your study according to the main results obtained.

Lines 297-299 are too speculative and should be delated. Moreover, lines 299-3030 are not conclusions of your study and look like an addition of lines 288-291…

Moreover, in the attached file I write more specific comments in the text, please see it 

Author Response

(The authors gave the same response as above.)

Round 2

Reviewer 1 Report

Re-Review of Article

General Comments

The authors made significant changes to improve both the content and clarity. The added paragraph in the introduction on laterality is appreciated. The authors should make a statement describing how mares were identified since pairs of horses performing allogrooming are the same. There are a few places that need spaces etc., but otherwise the formatting, tables and figures are well done.

Author Response

Thank you again for your review and feedback

Reviewer 3 Report

The paper improved a lot

Author Response

(The authors gave the same response as above.)
